# Recent Advances in Antiviral Activities of Triterpenoids

**DOI:** 10.3390/ph15101169

**Published:** 2022-09-21

**Authors:** Yue Liu, Liangyu Yang, Hong Wang, Yongai Xiong

**Affiliations:** 1Department of Pharmaceutics, Key Laboratory of Basic Pharmacology of Guizhou Province and School of Pharmacy, Zunyi Medical University, Zunyi 563000, China; 2Key Laboratory of Basic Pharmacology of Ministry of Education and Joint International Research Laboratory of Ethnomedicine of Ministry of Education, Zunyi Medical University, Zunyi 563000, China

**Keywords:** terpenoids, biological activity, natural products

## Abstract

Triterpenoids, important secondary plant metabolites made up of six isoprene units, are found widely in higher plants and are studied for their structural variety and wide range of bioactivities, including antiviral, antioxidant, anticancer, and anti-inflammatory properties. Numerous studies have demonstrated that different triterpenoids have the potential to behave as potential antiviral agents. The antiviral activities of triterpenoids and their derivatives are summarized in this review, with examples of oleanane, ursane, lupane, dammarane, lanostane, and cycloartane triterpenoids. We concentrated on the tetracyclic and pentacyclic triterpenoids in particular. Furthermore, the particular viral types and possible methods, such as anti-human immunodeficiency virus (HIV), anti-influenza virus, and anti-hepatitis virus, are presented in this article. This review gives an overview and a discussion of triterpenoids as potential antiviral agents.

## 1. Introduction

Viral infections are currently the main infectious disease worldwide [1], accounting for more than two thirds of infectious diseases [2]. Viruses that invade the human body can be divided into two categories. One is a group of viruses that are long-term parasites in the human body, including chickenpox, rubella, herpes, measles, smallpox, polio, Japanese encephalitis, mumps, cytomegalovirus (CMV), hepatitis B/C virus (HBV/HCV), dengue virus (DENV), human influenza virus, human immunodeficiency virus (HIV), and human papilloma virus (HPV). The other group consists of viruses that are long-term parasites in animals such as chickens, dogs, pigs, horses, and sheep, which are close to humans. These viruses, which can also infect humans across species, include avian influenza virus, rabies virus, hantavirus, and other viruses [3].

Viruses usually spread from one region to another, and in the past 10 years, the large-scale outbreaks of severe acute respiratory syndrome (SARS), Middle East respiratory syndrome (MERS), Ebola, and other viruses have caused considerable threats to human health. Moreover, the residual effects of viral infections cause considerable deleterious effects, pain, and inconvenience to patients. For example, the “cytokine storm” triggered by influenza virus infection can cause lung inflammation and acute respiratory distress syndrome, which is an important cause of mortality [4,5]. The neurotoxicity induced by herpes zoster virus infection often causes ganglion inflammation and necrosis. After invading the sensory nerve endings of the skin, this virus can move along the nerve to the ganglion of the posterior root of the spinal cord, where it remains latent.

Presently, vaccines and screening antiviral drugs are the main drugs used for the prevention and treatment of human viral infections [6]. However, it is difficult to develop an effective vaccine, and the side effects are unpredictable, such as the blood clots that can occur following the administration of the severe acute respiratory syndrome coronavirus-2 (SARS-CoV-2) vaccine [7]. In addition, vaccination is not effective in 100% of patients, and there is a chance that someone who is vaccinated against the coronavirus could still catch it [8]. Moreover, many viruses are zoonotic, and vaccines against human viral disease cannot eradicate animal-derived viruses. Presently, the focus of antiviral drug research is limited to several types of viruses, such as HIV, herpes, influenza, hepatitis, and respiratory viruses [9].

Triterpenoids are important plant metabolites characterized by six isoprene (2-methyl-1,3-butadiene, C_5_H_8_) units. Studies reported that triterpenoids have a wide range of pharmacological effects and important biological activities, especially anti-inflammatory, hypoglycemic, antitumor, and antiviral activities [10]. Triterpenoids have been proven to be important in the defense mechanisms of plants against pathogenic infection. Recently, an increasing number of studies have shown that triterpenoids exhibit excellent antiviral activities [11]. These compounds can prevent viral adsorption and invasion into host cells in the early stage and thereby inhibit the viral replication process after cell infection.

This review summarizes the antiviral activity of triterpenoids and the mechanism underlying these effects and mainly focuses on their mechanism of action against HIV, influenza virus, HBV, HCV, and coronavirus (CoVs) (Figure 1). This review provides important evidence to support the design and development of triterpenoids as antiviral drugs.

## 2. Structure and Classification of Triterpenoids

According to the number of isoprene units in the molecule, terpenoids can be divided into groups such as monoterpenes, sesquiterpenes, diterpenes, triterpenes, tetraterpenes, and polyterpenes. Among these groups, the tetracyclic triterpenes and pentacyclic triterpenes are the most common. Triterpenoids exist widely in nature in the free form or as glycosides or esters combined with sugars. There are numerous modifications of the carbon skeleton of triterpenoids, which is a complex and diverse structure that imparts a variety of functions to these compounds, making them the focus of considerable interest. Triterpenes are a major class of chemical compounds found in natural plants and can be categorized into acyclic, monocyclic, tricyclic, tetracyclic, and pentacyclic triterpenoids [10]. Among them, tetracyclic and pentacyclic triterpenoids have been the focus of more extensive attention than the other classes. Tetracyclic triterpenoids can be divided into dammarane, lanostane, tirucallane, cycloartane, cucurbitanes, and meliacane types. Pentacyclic triterpenoids can be divided into oleanane, ursane, lupine, and friedelane types [12].

## 3. Antiviral Activities and Mechanism

### 3.1. Anti-HIV

Acquired immunodeficiency syndrome (AIDS) is a disease of the cell-mediated immune system or T lymphocytes of the human body. In AIDS, the helper T-cell count is reduced, which directly stimulates the production of antibodies by B cells. Consequently, the body’s natural defense system against AIDS is destroyed. In addition, other immune cells such as monocyte-phagocytes, B lymphocytes, and natural killer (NK) cells are also damaged to varying degrees, which promotes the occurrence of various serious opportunistic infections and tumors [9]. The life cycle of HIV-1 includes adsorption, fusion, reverse transcription, integration, transcription, translation, and assembly.

Viral drug targets include reverse transcriptase (RT), integrase (IN), protease (PR), ribonuclease H (RNase H), envelope glycoprotein (gp120), and transmembrane protein (gp41). Although highly active antiretroviral therapy (HAART) can greatly reduce the morbidity and mortality of AIDS, it has serious side effects and is associated with induced drug resistance. Studies have identified triterpenoids with anti-HIV-1 activity, and most compounds can act on multiple key HIV enzymes (Table 1).

Glycyrrhizin (GL, **1**) (Figure 2), also known as glycyrrhizic acid, which is a pentacyclic triterpenoid component isolated from licorice root, *Glycyrrhiza radix*, effectively inhibits the replication of HIV [13,14]. GL (**1**) can increase the number of OKT4 lymphocytes in HIV-infected patients, showing a good inhibitory effect on the incubation period of the HIV virus [15]. GL (**1**) can inhibit HIV replication by reducing the activity of protein kinase C (PKC) [16] and inducing β chemokines [17].

C-C chemokine receptor 5 (CCR5) has a critical role as a receptor required for HIV-1 to penetrate cells, and its mRNA expression is regulated by C-C motif chemokine ligand 2 (CCL2) or interleukin (IL)-10. In the absence of these regulators, the expression of CCR5 mRNA is significantly reduced, which can effectively inhibit the entry of CCR5-mediated HIV into cells [9,18]. Studies have reported that GL (**1**) inhibited the production of CCL2 and IL-10 in infected neutrophils, thereby reducing HIV replication in macrophages [19] and inhibiting the CCL2- and IL-10-mediated expression of CCR5 [20]. Moreover, GL (**1**) also reduces the fluidity of the cell membranes and the cell fusion induced by HIV-1.

GL (**1**) and its metabolite 18β-glycyrrhetinic acid (18β-GA, **2**) (Figure 2 and Figure 3) stimulate macrophages to produce nitric oxide (NO) and induce the transactivation of the expression of inducible NO synthase (iNOS) through nuclear factor (NF)-kB [21]. In addition, the mechanism of the anti-HIV-1 activity of GL was proven to be related to the structure of the polyanion [22,23]. There are three carboxyls in the chemical structure of GL (**1**), which enables it to block the binding of HIV-1 with receptors on CD4+ cells and interfere with the activity of PKC to inhibit the adsorption of the virus to target cells [24]. In addition, the infectivity of HIV-1 was reduced after GL treatment [25].

Ursolic acid (UA, **10**) (Figure 2 and Figure 4) is a pentacyclic triterpenoid compound that is widely distributed in natural plants such as *Mimusops caffra*, *Ilex paraguarieni*, and *Glechoma hederacea* and has antiviral activity [26]. In 1996, Quere et al. [27] found that it inhibits HIV-1 protease. Studies in 2007 found that UA (**10**) and its acetylate acetylursolic acid (**11**) (Figure 2 and Figure 4) inhibit the activity of HIV-1 protease, with half-maximal inhibitory concentration (IC_50_) values of 8 and 13 μM, respectively [28]. Ursonic acid (UNA, **13**) (Figure 2 and Figure 4) is a naturally occurring pentacyclic triterpenoid that has been extracted from certain medicinal herbs such as *Ziziphus jujuba* Mill. UNA (**13**) is different from **10** because it has a keto group at the C-3 position, which increases its biological activities [29], and it has been reported to inhibit HIV-1 protease [30].

A variety of triterpenoids isolated from plants of the *Schisandra* family have been shown to have anti-HIV effects. The cycloartane triterpenoid lancifodilactone F (**43**) (Figure 2) was isolated from the leaves and stems of *Schisandra lancifolia* (Rehd. et Wils) A. C. Smith. Compound **43** showed anti-HIV activity with a half-maximal effective concentration (EC_50_) of 20.69 μg/mL and a selectivity index > 6.62 [31]. A trinorcycloartane triterpenoid, lancifodilactone H (**44**); an A-ring-secocycloartane triterpenoid, lancifoic acid A (**45**); and nigranoic acid (**46**) isolated from *S. lancifolia* can inhibit HIV, with EC_50_ values of 16.6, 16.2, and 10.3 μg/mL, respectively (Figure 2) [32]. The triterpenoid angustific acid A (**48**) (Figure 2), isolated from *Kadsura angustifolia*, exhibited inhibitory activity against HIV, with an EC_50_ value of 6.1 μg/mL and a therapeutic index (TI) > 32.8 [33]. The triterpenoid lancilactone C (**50**) (Figure 2), isolated from *Kadsura lancilimba*, showed inhibitory effects on the replication of HIV, with an EC_50_ value of 1.4 μg/mL and a TI of greater than 71.4 [34]. The cycloartane triterpenoids nigranoic acid (**46**) and kadsuranic acid A (**47**), isolated from *Kadsura heteroclita*, have significant inhibitory effects on HIV-1 PR, with IC_50_ values of 15.79 and 20.44 μM, respectively (Figure 2) [35].

Oleanolic acid (OA, **8**), betulinic acid (BA, **16**), betulin (**17**), and lupeol (**18**) are representative pentacyclic triterpenoids that are widely found in nature with a broad antiviral spectrum (Figure 2). Lupeol (**18**), extracted from *Hemidesmus indicus* R. Br., has an inhibitory effect on HIV-1 RT-related ribonuclease, with an IC_50_ of 11.6 μM [36]. BA (**16**) is a naturally occurring pentacyclic lupane-type triterpenoid that is usually isolated from birch trees, and it has potent anti-HIV activity (EC_50_ and IC_50_ of 1.4 μM and 13 μM, respectively) [37]. What differentiates these compounds from traditional anti-HIV drugs is that the acylated derivatives at position C-3 of BA (**16**) inhibit HIV maturation, and the amidated derivatives at position C-28 prevent HIV from integrating into cell membranes [38], as shown in Figure 5 [24]. Bifunctional BA (**16**) derivatives are the series of compounds obtained by modifying positions C-3 and C-28 at the same time. The strongest anti-HIV effect is observed in ([[N-[3beta-O-(3′,3′-dimethylsuccinyl)-lup-20(29)-en-28-oyl]-7-aminoheptyl]-carbamoyl] methane), with an EC_50_ value of 0.0026 μM [39]. BA (**16**) and its derivatives with modifications at both C3-OH and C17-COOH can block virus fusion and virus maturation simultaneously as bifunctional inhibitors [40].

The C-3 position of BA (**16**) has been modified in studies, and 3-O-(3′,3′-dimethylsuccinyl)-betulinic acid (PA-457, **19**) (Figure 2 and Figure 6), also known as bevirimat (BVM), has been screened as the first compound in the class of HIV maturation inhibitors (MIs) [41,42], with an IC_50_ value of 10.3 nM [43]. In addition, the disadvantages of BVM (**19**) include its poor solubility in aqueous and biological organic media as well as its high plasma protein binding [38]. A series of phosphate derivatives of 3-carboxyacylbetulin, which had significant anti-HIV activity, with IC_50_ values of 0.02–0.22 µM in vitro, have been synthesized [44]. The inhibitory effect of compound **20** (Figure 2 and Figure 6) (IC_50_ = 0.02 µM, TI = 1250) on viral replication is equivalent to that of the positive control BVM (**19**) (IC_50_ = 0.03 µM, TI = 967), but it had better selectivity. The capsid protein (CA) CTD-SP1 might be the target of compound **20** against HIV.

Betulin (**17**) is a natural pentacyclic lupane-structure triterpenoid that possesses a wide range of pharmacological activities and exhibited the highest inhibitory activity against HIV-1 integrase, with an IC_50_ of 17.7 μM, followed by BA (**16**) with an IC_50_ value of 24.8 μM. OA (**8**) and UA (**10**) showed moderate activity, with IC_50_ values of 30.3 μM and 35.0 μM, respectively [45]. In addition, OA (**8**) from olives was found to completely inhibit HIV-1 protease inhibition, with an IC_50_ value of 57.7 μM [46].

Some triterpenoids isolated from *Ganoderma lucidum* and *Ganoderma sinense* showed inhibitory activity against the HIV-1 virus and protease. Ganodermanontriol (**34**) and ganoderiol F (**35**) isolated from *G. lucidum* inhibited the pathological effects of HIV-PR on MT24 cells with IC_50_ values > 1.0 mM and 7.8 μg/mL, respectively (Figure 2) [47]. Furthermore, HIV-1 protease inhibition experiments showed that ganoderic acid B (**39**) and ganodriol B (**36**) in *G. sinense* inhibited HIV-1 protease (Figure 2) [47]. Ganoderiol F (**35**), ganoderic acid GS-2 (**40**), 20 (21)-dehydrolucidenic acid N (**41**), and 20-hydroxylucidenic acid N (**42**), isolated from *G. lucidum* were proven to inhibit HIV-1 protease, with IC_50_ values of 22, 25, 30, and 48 μM (Figure 2) [48].

Several tetracyclic triterpenoids isolated from ginseng [49], such as ginsenoside Rb1 (G-Rb1, **29**), ginsenoside Rb2 (G-Rb2, **30**), ginsenoside Rg3 (G-Rg3, **31**), ginsenoside Rg1 (G-Rg1, **32**), and ginsenoside Rh1 (G-Rh1, **33**), showed strong inhibitory activity against HIV (Figure 2). G-Rb1 (**29**) eliminated HIV-1-infecting macrophages through the AKT pathway, whereas G-Rh1 (**33**) mainly eliminated HIV-1 infection by inhibiting the pyruvate dehydrogenase kinase 1 (PDK1)/Akt pathway [50,51].

Compounds **21**–**23** (Figure 2) are lupane-type pentacyclic triterpenoids, isolated from *Cassine xylocarpa* and *Maytenus cuzcoina*, that showed obvious effects against HIV in a lymphoblastoid cell line (MT-2), with IC_50_ values of 4.08, 4.18, and 1.70 μM, respectively [52]. In recent years, new anti-HIV triterpenoid compounds have constantly been discovered. Compound 53 (Figure 2) was isolated from *Antirhea chinensis* and displayed selective anti-HIV activity at 1.24 μM concentrations [53]. Compound 54 (Figure 2) was isolated from *Kleinhovia hospita* and exhibited sub-micromolar anti-HIV activity [54].

**Table 1 pharmaceuticals-15-01169-t001:** Summary of anti-HIV triterpenoids and their targets and activities.

Compounds	IC_50_	EC_50_	Targets	References
**1**	-	-	Entry inhibitors	[16]
**8**	57.7 μM 30.3 μM	-	Protease inhibitors Integrase inhibitors	[46] [45]
**10**	8 μM	-	Protease inhibitors	[28]
	35 μM	-	Integrase inhibitors	[45]
**11**	13 μM	-	Protease inhibitors	[28]
**13**	-	-	Protease inhibitors	[30]
**16**	24.8 μM	-	Integrase inhibitors	[45]
**17**	17.7 μM	-	Integrase inhibitors	[45]
**18**	11.6 μM	-	Reverse transcriptase inhibitor	[36]
**19**	0.03 μM	-	Maturation inhibitors	[44]
**20**	0.02 μM	-	-	[44]
**21**	4.08 μM	-	Replication inhibition	[52]
**22**	4.18 μM	-	Replication inhibitors	[52]
**23**	1.70 μM	-	Replication inhibitors	[52]
**34**	>1.0 mM	-	Protease inhibitors	[47]
**35**	7.8 μg/mL	-	Protease inhibitors	[48]
**40**	30 μM	-	Protease inhibitors	[48]
**41**	48 μM	-	Protease inhibitors	[48]
**42**	25 μM	-	Protease inhibitors	[48]
**43**	-	20.69 μg/mL	-	[32]
**44**	-	16.6 μg/mL	-	[32]
**45**	-	16.2 μg/mL	-	[32]
**46**	-	10.3 μg/mL	-	[32]
	15.79 μM	-	Protease inhibitors	[35]
**47**	20.44 μM	-	Protease inhibitors	[35]
**48**	-	6.1 μg/mL	-	[33]
**50**	-	1.4 μg/mL	-	[34]
**53**	-	1.24 μM	-	[53]
**54**	-	-	-	[54]

### 3.2. Anti-Influenza Virus

The influenza virus is the respiratory virus posing the greatest threat to human health, and it causes millions of people to have upper respiratory tract infections yearly. The high mutation rate of influenza viruses and the recombination between viruses makes it difficult for anti-influenza drugs to effectively treat infections caused by the changing influenza viruses. Currently, the effectiveness of the two types of chemical drugs, M2 ion channel blockers and neuraminidase inhibitors, for the treatment and prevention of influenza viruses is limited by severe drug resistance. Although vaccines are the key drugs for the prevention and treatment of viral infections in humans, some viruses such as CoV are zoonotic, and vaccines against humans cannot be used to eradicate them. In addition, the extremely high gene reconfiguration efficiency of influenza viruses makes them effective targets for vaccines.

Triterpenes have also shown obvious effects against the influenza virus, such as ursolic acid and betulinic acid derivatives, which exhibit significant anti-influenza-virus activity (Table 2) [40]. GL (**1**) inhibits the replication of the human parainfluenza virus type 2 (HPIV2) by inhibiting the synthesis of genomic RNA, mRNA, and protein [55]. Furthermore, the combination of GL (**1**) and ribavirin effectively treated H1N1 influenza virus infection in vivo and inhibited the production of pro-inflammatory cytokines IL-6, tumor necrosis factor (TNF)-α, and IL-1β induced by viral infection, indicating that GL (**1**) reduced inflammation and inhibited the viral infection [56].

Studies found that a series of glycyrrhetinic acid (GA, **3**) (Figure 2) derivatives and cyclodextrin (CD) sowed good anti-A/WSN/33 (H1N1) influenza virus activity [57]. The oleanane-type triterpene echinocystic acid (EA, **9**) (Figure 2) and galactose conjugate inhibited the H1N1 influenza virus [58] and significantly reduced the cytopathic effect (CPE) on Madin–Darby canine kidney (MDCK) cells, with an EC_50_ of 5 μM. Subsequently, replacing EA (**9**) with OA (**8**) or UA (**10**) greatly improved the anti-influenza-virus activity of the compound, and compounds Y3 and Y5, obtained after the partial acetylation of galactose, showed the best anti-influenza-virus activity, with EC_50_ values of 14.2 and 15.1 μM, respectively [55,58].

The structural modification of UA (**10**) produced a compound that prevented the H5N1 influenza virus from penetrating host cells [59]. The esterification of the C-17 carboxyl group of UA (**10**) significantly increased the antiviral activity, whereas the conversion of the carboxyl group into an amide increased the antiviral activity and reduced the cytotoxicity.

Diammonium glycyrrhizin (DG, **4**) (Figure 2) is a salt form of glycyrrhizinate, a major active component of licorice root extract, with various pharmacological activities that is widely used as an anti-inflammatory. DG (**4**) has antiviral effects, reduces inflammation induced by viral infections, exhibits anti-H1N1 activity [60], and significantly reduces the virus titer and nucleoprotein expression of H1N1-infected MDCK cells. Furthermore, DG (**4**) does not directly inactivate the H1N1 virus or act on the adsorption and penetration of the virus, but it upregulates the expression of the interferon-γ (IFN-γ) gene in H1N1-infected cells. Moreover, DG (**4**) downregulated the expression of the TNF-α gene, reducing the inflammatory response induced by H1N1 and the immune damage to host cells. G-Rb1 (**29**) interacts with viral hemagglutinin (HA) protein to prevent viral adhesion to the α2-3 sialic acid receptor on the surface of the host cell and thereby interferes with the adhesion process of the virus [61].

The compound CYY1-11, which was selected from a mini-library of pentacyclic triterpene–cyclodextrin conjugates, is a hexavalent BA (**16**) derivative with good activity against the H1N1 influenza virus, with an IC_50_ of 5.2 μM. CYY1-11 directly targets the influenza HA protein, inhibiting viral replication and preventing viral particles from adsorbing and infecting host cell receptors [62]. Paracaseolin A (**24**) (Figure 2) is a triterpenoid paracaseolin isolated from *Sonneratia paracaseolaris* that shows a significant anti-H1N1-virus effect, with an IC_50_ of 28.4 μg/mL [63].

**Table 2 pharmaceuticals-15-01169-t002:** Summary of anti-influenza-virus triterpenoids and their targets and activities.

Compounds	IC_50_	EC_50_	Targets	References
**1**	-	-	Replication inhibitors	[55]
**4**	-	-	Immunomodulator	[60]
**9**	-	5 μM	Entry inhibitors	[58]
**10**	-	-	Entry inhibitors	[59]
**24**	28.4 μg/mL	-	-	[63]

### 3.3. Anti-CoV

Since the beginning of the 21st century, outbreaks of CoV have caused fatal pneumonia in humans. CoV is a large family of viruses, and in only two decades, three strains have crossed species barriers, rapidly emerging as human pathogens that have resulted in life-threatening diseases with pandemic potential, such as SARS-CoV, MERS-CoV, and SARS-CoV-2 [64]. SARS-CoV and MERS-CoV are zoonotic viruses derived from bats and dromedaries [65], and the outbreak of SARS-CoV-2 at the end of 2019 caused considerable human mortality and large-scale sustained illness [66]. SARS-CoV-2 is considered a respiratory disease, but studies have shown that the virus affects multiple organs, including the central nervous system [67]. The high incidence of CoVs poses a serious threat to human health and creates considerable challenges to human life.

GL (**1**) inhibits SARS-CoV replication, adsorption, and penetration, with a better antiviral activity than that of ribavirin and an SI of 67 [68]. In a previous study, GL (**1**) was found to be most effective when administered both during and after the adsorption period. Furthermore, GL (**1**) induced NOS production in Vero cells and inhibited viral replication when a nitrous oxide donor (DETA NONOate) was added to the culture medium [68].

Studies testing the activity of 15 GL (**1**) derivatives found that the introduction of 2-acetamido-beta-d-glucopyranosylamine into the glycoside chain of GL (**1**) increased the anti-SARS-CoV activity by 10-fold compared to that of the parent GL (**1**) [69]. Amides and conjugates of GL (**1**) with two amino acid residues and a free 30-COOH function were found to exhibit up to 70-fold increased activity against SARS-CoV compared to the parent compounds; however, the accompanied increased cytotoxicity decreased the selectivity index [69].

GL (**1**) can be used for the treatment and prevention of SARS-CoV-2 [70], and there are numerous possible mechanisms mediating its activity against viruses, including increased NO production in macrophages, the regulation of transcription factors and cellular signaling pathways, the direct alteration of the viral lipid bilayer membrane, and binding to the angiotensin-converting enzyme 2 (ACE2) receptor [70,71,72]. Furthermore, GL (**1**) inhibited the protease M of SARS-CoV-2, which can process polyproteins, the translation products of viral RNA, which are attractive drug targets (Table 3) [73].

Some active compounds such as GL (**1**) strongly bind to part of the amino acid residue of protease M, which can inhibit the enzymes, and they have good solubility, absorption, and permeability and are nontoxic, and noncarcinogenic. Magnesium isoglycyrrhizinate (MgIG, **7**) (Figure 2), a magnesium salt of the 18-α glycyrrhizic acid stereoisomer, has been identified as a potential adjuvant treatment for SARS-CoV-2 (Table 3) [74]. OA (**8**), which also exerts anti-SARS-CoV-2 activity [75], is a 3CL hydrolase inhibitor of this virus, and the binding energy consumed by 3CL hydrolase is similar to that of radcivir but lower than that of ritonavir. Studies have shown that maslinic acids (MA, **51**), UA (**10**), BA (**16**), and betulin (**17**) all have good inhibitory effects on SARS-CoV-2 M^pro^, with IC_50_ values of 3.22, 12.57, 14.55 and 89.67 μM, respectively [76]. Compound **52**, a derivative of MA (**51**), can inactivate the SARS-CoV-2 virus and inhibit the adsorption and replication of the virus (IC_50_ = 4.12 μM). It also has a good inhibitory effect on MERS-CoV (IC_50_ = 6.25 μM) [77] (Table 3).

**Table 3 pharmaceuticals-15-01169-t003:** Summary of anti-CoV triterpenoids and their targets and activities.

Compounds	IC_50_	Targets	Virus	References
**1**		Inhibits SARS-CoV replication, adsorption, and penetration;	SARS-CoV	[68]
		Protease inhibitor	SARS-CoV-2	[73]
**7**		-	SARS-CoV-2	[74]
**8**		3CL hydrolase inhibitor	SARS-CoV-2	[75]
**10**	12.57	SARS-CoV-2 M^pro^ inhibitor	SARS-CoV-2	[76]
**16**	14.55	SARS-CoV-2 M^pro^ inhibitor	SARS-CoV-2	[76]
**17**	89.67	SARS-CoV-2 M^pro^ inhibitor	SARS-CoV-2	[76]
**51**	3.22	SARS-CoV-2 M^pro^ inhibitor	SARS-CoV-2	[76]
**52**	4.12	Inhibits SARS-CoV replication and adsorption	SARS-CoV	[77]
	6.25	-	MERS-CoV	[77]

### 3.4. Anti-Hepatitis Virus

Viral hepatitis can be caused by a variety of hepatitis viruses, which are usually divided into A, B, C, D, and E types. The A and E forms of the virus cause acute infections and can be transmitted through the fecal–oral route. The B, C, and D viruses mainly cause chronic infections, which are transmitted through blood and body fluids. Recently, traditional Chinese medicine (TCM) practices and formulations have been widely explored for the treatment of viral hepatitis and, in particular, the pentacyclic triterpenes have been a focus.

#### 3.4.1. Anti-Hepatitis A Virus (HAV)

The hepatitis A virus (HAV), which is the main cause of hepatitis A, is a food-borne disease, and vaccination is the most effective preventative treatment. Early research has shown that GL (**1**) achieved a concentration-dependent inhibition of the replication of HAV in PLC/PRF/5 cells (Table 4) [78]. Gi-Rb1 (**29**) and G-Rg1 (**32**) have been shown to significantly inhibit HAV in vitro (Table 4) [79].

#### 3.4.2. Anti-HBV

Chronic hepatitis B is a long-term progressive disease caused by HBV infection, which leads to hepatitis, cirrhosis, and even liver cancer. Presently, the main drugs for the treatment of chronic hepatitis B are IFN in combination with nucleoside drugs such as lamivudine, adefovir dipivoxil, and clevudine. The low treatment success rate and severe side effects of IFN and the increased resistance to nucleoside drugs has limited the clinical application of these drugs [80]. Triterpenoids have been proven to have obvious advantages in the treatment of viral diseases (Table 5).

A study showed that lamivudine in combination with GL (**1**) effectively controlled HBV replication in chronic HBV carriers [81]. Studies have led to the isolation of three new friedelolactones, violaic A (**26**), violaic B (**27**), and violalide (**28**), from *Viola diffusa Ging*, which block the secretion of hepatitis B surface antigen (HBsAg), with IC_50_ values of 26.2, 33.7, and 104.0 μM, and block the secretion of hepatitis B e antigen (HBeAg), with IC_50_ values of 8.0, 15.2, and 21.6 μM (Figure 2) [82]. Cleavage-A-pentacyclic triterpene derivatives have anti-HBV activity. Compared with the parent compound, OA (**8**), the inhibitory effects of compounds OA-2-2BV and OA-4 on HBeAg secretion were increased 1000-fold [83]. GL-3, a derivative of GL (**1**), showed an inhibitory activity of 91.56% on HBV DNA, which is equivalent to that of lamivudine. Astragaloside IV (AS-IV, **49**) (Figure 2) is an effective component of the TCM formulation *Astragalus membranaceus*. AS-IV (**49**) showed a good anti-HBV effect by inhibiting the cell proliferation or viral DNA replication of infected cells [84]. G-Rg3 (**31**) inhibits DNA replication and viral protein formation [85], and G-Rg1 (**32**) enhanced the immunoactivity of an anti-HBsAg vaccine, thereby promoting the secretion of IgG2b and IFN-γ by Th1 cells in mice and the secretion of IgG1 and IL-4 from Th2 cells to exert an anti-HBV effect [86].

**Table 5 pharmaceuticals-15-01169-t005:** Summary of anti-HBV triterpenoids and their targets and activities.

Compounds	IC_50_	Targets	Cells	References
**1**	-	Replication inhibitors	-	[81]
**26; 27; 28**	HBsAg: 26.2, 33.7, and 104.0 μM HBeAg: 8.0, 15.2, and 21.6 μM	Inhibits the secretion of HBsAg and HBeAg	HepG2.2.15 cells	[82]
**31**	-	Replication inhibitors	-	[85]
**32**	-	Immunomodulator	-	[86]
**49**	HBsAg: 28.15 μM HBeAg: 33.38 μM	Replication inhibitors	HepG2.2.15 cells	[84]

#### 3.4.3. Anti-HCV

HCV infection usually causes chronic hepatitis C (CHC), liver cirrhosis, and liver cancer, and approximately 185 million people are infected with the HCV. During the HCV replication process, NS3/4A serine protease mediates the cleavage of polyprotein precursors, which promotes the maturation and function of nonstructural proteins. Clinically, >95% of CHC patients are treated with antiviral drugs that act against NS3/4A protease, which can induce a sustained virological response. However, some patients with liver cirrhosis still develop hepatocellular carcinoma, and there is serious drug resistance to these agents, which have multiple side effects. Triterpenoids have been proven to have obvious advantages in the treatment of HCV (Table 6).

The active ingredient, OA (**8**), was isolated from *Dipsacus asperoides* and exhibited a good anti-HCV activity [87]. The structure–activity relationships of OA (**8**) against HCV are shown in Figure 7. [24]. EA (**9**), which is the derivative of OA (**8**), also showed activity against HCV, and the mechanism of action may be related to inhibiting the binding of the HCV envelope protein E2 to the host cell CD81 receptor, which prevents HCV from recognizing the host. The A and B loops and the left side structures of the C and E loops of EA (**9**) are highly conserved, and consequently its modification has an adverse effect on the antiviral activity, whereas a reasonable modification of the D loop improves antiviral activity [24]. To improve the water solubility of OA (**8**) and EA (**9**), they were coupled with β-CD to obtain a series of new triazole-bridged α-CD-pentacyclic triterpenoids. Studies have shown that the new conjugate has reduced hydrophobicity and no cytotoxicity or hemolytic activity [88].

The NS5b gene plays a key role in the synthesis and replication of viral RNA, catalyzing the synthesis of negative RNA strands using HCV genomes as templates and the synthesis of negative RNA strands with HCV genomes as templates [89]. OA (**8**) and UA (**10**) competitively inhibit the activities of NS5B and RNA polymerase, and the IC_50_ values of OA (**8**) and UA (**10**) were about 1.8 μM and 6.8 μM, respectively [89]. HCV infection can cause changes in mitochondrial dynamics and weaken the immunity. G-Rg3 (**31**) inhibited the proliferation of HCV-infected cells by affecting their expression of p21, repairing abnormal mitochondrial division, and inhibiting persistent HCV infection [90].

**Table 6 pharmaceuticals-15-01169-t006:** Summary of anti-HCV triterpenoids and their targets and activities.

Compounds	IC_50_	Targets	Cells	References
**8**	1.8 μM	Entry inhibitors Protease inhibitors and polymerase inhibitors	Hep G2-5B cells	[87,89]
**9**	-	Entry inhibitors	-	[24]
**10**	6.8 μM	Polymerase inhibitors	Hep G2-5B cells	[89]
**31**	-	Infection inhibitors	Human hepatoma Huh7 and Huh7.5.1 cells	[90]

### 3.5. Anti-Rotavirus (RV)

RV is the main cause of severe enteritis in children under 5 years old, and it mainly infects the intestinal mucosa cells and induces enterotoxins, leading to malnutrition and even death in children. There is currently no clinical treatment for RV infection. 2α-Hydroxyursolic acid (**12**) (Figure 2) extracted from guava leaves has a strong anti-human-rotavirus (HRV) effect in vitro, which stimulates the secretion of IFN-γ, enhancing immune function (Table 7) [91]. UA (**10**) was also reported to inhibit the replication of RV and affect the maturation of viral particles (Table 7) [92]. Moreover, G-Rb2 (**30**) and 20(S)-GRg3 (**31**) exhibited good anti-RV effects as immunomodulators (Table 7) [93].

### 3.6. Anti-Herpesvirus

Herpesviruses are double-stranded DNA viruses with icosahedral symmetry, belonging to the Herpesviridae family. Most herpes viruses can cause human infections, including herpes simplex virus (HSV)-1 and HSV-2, the African lymphocyte Epstein–Barr virus (EBV), Kaposi’s sarcoma-associated herpesvirus (KSHV), and human cytomegalovirus (HCMV) [94]. The herpesvirus can infect any organ or tissue in the human body to cause a variety of diseases. Recently, herpesvirus infection has become one of the main causes of death in people with impaired immune function. Presently, there are still no anti-herpesvirus drugs with high efficiency and specificity and without side effects.

Carbenoxolone sodium (CBX, **5**) and cicloxolone sodium (CCX, **6**) are derivatives of triterpenoid GL (**1**), found in glycyrrhiza plants, that inhibit HSV-1 and HSV-2 (Figure 2). These compounds significantly inhibit viral replication and thereby reduce the number of infectious viral particles (Table 8) [95]. Another study showed that GL (**1**) effectively inhibits the adsorption of HSV to rat brain capillary endothelial cells and polymorphonuclear leukocytes, thereby significantly reducing the inflammatory response caused by the HSV infection of host cells [96]. GL (**1**) induces antiviral effects through the prevention of the early invasion of host cells by EBV, with an IC_50_ of 0.04 mM (Table 8) [97]. GL (**1**) is a strong beclin 1 activator of autophagy, which inhibits the replication of the virus [98]. Moreover, combining GL (**1**) with some antiviral and anti-inflammatory enzymes or proteins such as lysozyme (LYS) and lactotransferrin (LAC) produced a synergistic effect [99] against the activity of the HSV-1 virus. Furthermore, the combination can reduce the adverse reactions such as edema, hypokalemia, and hypertension caused by large doses of GL (**1**). Generally, drugs have little effect on the viral incubation period, but GL (**1**) inhibits the replication of latent KSHV in B lymphocytes [100]. Furthermore, GL (**1**) reduces the synthesis of KSHV latency protein, inducing the apoptosis of infected cells, and thereby exerts antiviral effects [101]. 20(S)-GRg3 (**31**) has a significant inhibitory effect on HSV-1 and HSV-2, with an IC_50_ of 35 μM (Table 8) [102].

HCMV is a ubiquitous herpes virus that can latently infect hosts throughout their life cycle and is reactivated when the immune system is weak. Currently, there is no vaccine for HCMV infection and existing antiviral drugs mainly target viral enzymes and have obvious adverse reactions [103]. Therefore, it extremely imperative to find compounds with anti-HCMV properties. OA (**8**) and UA (**10**) have anti-guinea pig cytomegalovirus (GPCMV) effects [104,105] with inhibitory rates of 18%, 35%, and 84% on GPCMV adsorption, penetration, and replication, respectively. Furthermore, the antiviral target is the inhibition of virus synthesis.

**Table 8 pharmaceuticals-15-01169-t008:** Summary of anti-herpes-virus triterpenoids and their targets and activities.

Compounds	IC_50_	Targets	Virus	References
**1**	-	Immunomodulator Entry inhibitors Replication inhibitors	HSV EBV HSV-1, KSHV	[96] [97] [98,100]
	0.04 mM			
	-			
**5; 6**	-	Replication inhibitors	HSV-1, HSV-2	[95]
**31**	35 μM	-	HSV-1, HSV-2	[102]

### 3.7. Anti-DENV

DENV is the most common arboviral disease worldwide, with 100 million symptomatic cases annually [106], and despite its major impact on global human health and the huge economic burden, there is no antiviral drug available to treat the disease [107]. DENV was listed as one of the top 10 global health threats in 2019 by the World Health Organization (WHO) [108]. The first tetravalent DENV vaccine was licensed in 2015 for individuals aged 9 to 45 years; however, most cases are reported in infants and young children. These facts, coupled with the limited efficacy of the vaccine against DENV serotype 2, stress the need to continue the search for compounds with anti-DENV activity.

GL (**1**) has anti-DENV-2-virus activity (IC_50_ = 8.1 μM), with a low cytotoxicity on Vero E6 cells [13], which can inhibit DENV-2-induced cytopathic changes and reduce DENV-2 infectivity (Table 9). A structural modification of GL (**1**) showed that its conjugation with isoleucine and 11-aminoundecanoic acid produced potent anti-DENV-2 inhibitors with IC_50_ values of 1.2–1.3 μM [13].

CBX (**5**) has anti-DENV activity, and studies where DENV was incubated with CBX (**5**) showed a reduction in the viral titer and infectivity [109]. In addition, CBX (**5**) has a significant inhibitory effect on DENV RNA (Table 9). BA (**16**) inhibits DENV viral RNA synthesis and protein production, and it can also resist other mosquito-borne RNA viruses, such as Zika virus and Chikungunya virus (CHIKV), which usually spread with DENV [107]. BA (**16**), betulin (**17**), and betulinic aldehyde (**25**) (Figure 2) showed anti-DENV activity, with IC_50_ values of 4.3 ± 3.1, 4.1 ± 0.4, and 7.5 ± 1.1 μM (Table 9) [110]. In addition, ganodermanontriol (**34**) extracted from *G. lucidum* showed good inhibitory activity against DENV virus NS2B-NS3 protease (Table 9) [111].

**Table 9 pharmaceuticals-15-01169-t009:** Summary of anti-DENV triterpenoids and their targets and activities.

Compounds	IC_50_	Targets	References
**1**	8.1 μM	Inhibited DENV2-induced CPE and reduced DENV-2 infectivity	[13]
**5**	-	Virucidal effect	[109]
**16**	4.3 ± 3.1 μM	Replication inhibitors	[110]
**17**	4.1 ± 0.4 μM	Replication inhibitors	[110]
**25**	7.5 ± 1.1 μM	Replication inhibitors	[110]
**34**	-	Protease inhibitors	[111]

### 3.8. Anti-Norovirus (NoV)

NoV is the main pathogen causing nonbacterial acute gastroenteritis in humans and acute diarrhea in infants and young children [112]. NoV constitutes a considerable health risk to society and is the cause of >50% of viral gastroenteritis cases worldwide, whereas >85% of nonbacterial gastroenteritis cases are related to NoV [113]. Recently, considerable progress has been achieved in the development of anti-NoV drugs, and discovering compounds with anti-NoV activity from natural products is a laudable strategy. Astragaloside has anti-NoV effects [114] and has been proven to significantly downregulate the levels of TNF-α, IL-1β, and IL-6 while enhancing the anti-inflammatory responses of patients. Furthermore, it enhances the activity of the intestinal immune cells to eliminate viruses by regulating the protein kinase-R (PKR)/p-PKR signal transduction pathway.

### 3.9. Anti-Coxsackievirus (CV)

CV is divided into A and B types, and it is the main pathogen causing viral myocarditis and dilated cardiomyopathy in children and adolescents. The early innate immune response of the host to CV infection depends on IFN, and AS-IV (**49**) has been shown to upregulate the expression of IFN-γ mRNA, which inhibits CV [115]. GL (**1**) treatment has a significant effect on viral myocarditis caused by CV [115], and it inhibits the enhanced expression of TNF-α, IL-1β, IL-6, and other inflammatory cytokines caused by CV infection.

### 3.10. Anti-Adenovirus (AdV)

AdV was first isolated in 1949 when a biological sample from an organism with bovine nodular dermatitis disease was inoculated into chicken embryos. In 1953, human AdV (HAdV) was isolated from the tonsils of children. HAdV include seven types and 65 subtypes from A to G, which can cause diseases such as acute upper and lower respiratory tract infections, explosive conjunctivitis, gastroenteritis in infants, and acute hemorrhagic cystitis. 

AdV is highly contagious, especially in densely populated places and can be spread through droplets and contact, which leads to regional epidemics or outbreaks. HAdV3 is the main pathogen that causes acute respiratory diseases and pharyngeal conjunctival fever. Currently, there is no effective preventative vaccine, and specific drugs have not yet been discovered or developed. AS-IV (**49**) has anti-AdV activity in vitro [116] and damages the self-assembly stage of AdV by a mechanism of action that is related to the inhibition of the expression of hexon protein. In addition, this agent inhibits the replication and adsorption of AdV.

### 3.11. Anti-Enterovirus (EV71)

EV71 is one of the common pathogens responsible for hand, foot, and mouth disease (HFMD) [117]. Children who experience HFMD commonly have myocardial and nervous system damage, which causes systemic complications and seriously endangers the health of children. GL (**1**) inhibits EV71 infection of host cells and has been shown to have no direct inactivation effect on the EV71 virus but acts after the virus enters the cell to exert its antiviral effect [87].

G-Rb1 (**29**) exerts antiviral activity against EV71-infected suckling mice [118] and reduced the CPE of RD cells infected by EV71. In addition, G-Rb1 (**29**) reduces EV71-induced viral protein-1 expression both in vivo and in vitro while inducing natural immune responses in the body and enhancing IFN responses [118]. UA (**10**) showed strong inhibitory activity against EV71, and OA (**8**), asiatic acid (AA, **14**), and synthetic derivatives of GL (**1**) and OA (**8**) also showed inhibitory effects (Figure 2). These compounds could be further optimized as candidates for the treatment of EV71 [119]. The triterpenoids lanosta-7, 9 (11), 24-trien-3-one, 15;26-dihydroxy (GLTA, **37**), and ganoderic acid Y (GLTB, **38**), isolated from *G. lucidum,* showed anti-EV71 activity (Figure 2) [120]. Furthermore, GLTA (**37**) and GLTB (**38**) were proven to block the adsorption of viruses to host cells by interacting with viral particles (Table 10).

### 3.12. Anti-Porcine Reproductive and Respiratory Syndrome Virus (PRRSV)

Porcine reproductive and respiratory syndrome virus (PRRSV) is an enveloped single-stranded positive-stranded RNA virus, which constitutes a considerable threat to pigs. Current existing vaccines cannot provide comprehensive protection against this virus. GL (**1**) inhibits the cellular penetration of PRRSV and significantly reduces its proliferation and the expression of its encoded protein [121]. Gly-CDs synthesized based on GL (**1**) inhibit the invasion and replication of PRRSV by stimulating the antiviral innate immune response and inhibiting the accumulation of intracellular reactive oxygen species (ROS) caused by PRRSV infection (Table 11) [122]. Gly-CDs also stimulate cells to regulate the expression of some host restriction factors to inhibit PRRSV proliferation, including DDX53 and NOS3.

G-Rg1 (**32**) inhibited PRRSV infection in MARC-145 cells and porcine alveolar macrophages [123] and affects multiple stages of PRRSV adsorption, replication, and release. Furthermore, G-Rg1 (**32**) showed extensive inhibitory activity against PRRSV-2, including the highly pathogenic (HP)-PRRSV XH-gD and JXA1, NADC-30, and the classic strain VR2332. G-Rg1 (**32**) reduced the expression of the pro-inflammatory cytokines IL-1β, IL-8, IL-6, and TNF-α and reduced the activation of NF-κB signaling triggered by PRRSV infection (Table 11). A derivative of UA (**10**), compound **15** (Figure 2), directly inactivated PRRSV viral particles, thereby inhibiting replication to exert an antiviral effect (Table 11) [124].

**Table 11 pharmaceuticals-15-01169-t011:** Summary of anti-PRRSV triterpenoids and their targets and activities.

Compounds	Targets	References
**1; 15**	Replication inhibitors	[121,124]
**32**	Entry inhibitors Immunomodulator	[123]

### 3.13. Anti-Human Respiratory Syncytial Virus (HRSV)

HRSV is the main pathogen that causes bronchiolitis and pneumonia in infants and young children. However, there is currently no vaccine to prevent HRSV infection [125]. 18β-GA (**2**) inhibits HRSV adsorption and internalization and the stimulation of IFN secretion [126]. However, GL (**1**), isolated from licorice, exhibited no obvious inhibitory effect on HRSV.

## 4. Conclusions and Future Prospective

The rapid mutation of viruses enables them to easily develop resistance to drugs, which limits the use of currently available drugs. Therefore, the development of broad-spectrum antiviral drugs is important, especially for newly emerging viruses, such as SARS-CoV-2. Traditional herbal medicines and plant-based natural compounds are rich resources of new antiviral drugs. Many traditional herbal medicines possess antiviral activities against a plethora of viral strains, exerting their antiviral activities on the virus life cycle, including viral entry, replication, assembly, and release as well as the virus–host-specific interactions, allowing them to theoretically lead to the discovery of broad-spectrum antivirals.

Triterpenoids are the focus of considerable attention because of their novel chemical structure and unique mechanism of action. In recent years, an increasing number of studies have reported on the effects of triterpenoids on viruses. This present article systematically summarizes the recent advances in the study of the types, structural characteristics, and potential antiviral mechanisms of triterpenoids. The investigation of the antiviral mechanisms of these triterpenoids leads to the conclusion that they are mainly exerted through the inactivation of viral particles, the prevention of viral replication and entry into host cells, and the regulation of immunity. The present review aims to provide evidence to support the effectiveness of the strategy of discovering new antiviral drugs from triterpenoids.

However, there are still some obstacles to developing broad-spectrum antiviral drugs from triterpenoid compounds. The targets and mechanisms underpinning some natural products remain unclear, potential side effects have not been fully investigated, and the activities of triterpenoids are usually moderate. Advanced technology for chemical modification, investigations into underlying biological mechanisms, and progress in drug design will facilitate future developments in the field. In addition, the structure of triterpenoids determines that they have high polarity and poor bioavailability, which limits their biofilm permeability and absorption. Therefore, it is necessary to improve the dissolution and absorption ability of triterpenoids by chemical or pharmaceutical methods. For example, the esterification or amidation structure modification of triterpenoids can increase their lipid solubility, membrane permeability, and oral absorption rate. In addition, new pharmaceutical dosage forms and new technologies can also improve the oral bioavailability of triterpenoids, such as oral self-emulsifying systems, liposomes, microspheres, microcapsules, nanomicelles, etc. Furthermore, appropriate administration routes can also improve the bioavailability of triterpenoids, for example, transdermal drug delivery systems, colonic administration, nasal mucosa administration, pulmonary inhalation administration, etc.

## Figures and Tables

**Figure 1 pharmaceuticals-15-01169-f001:**
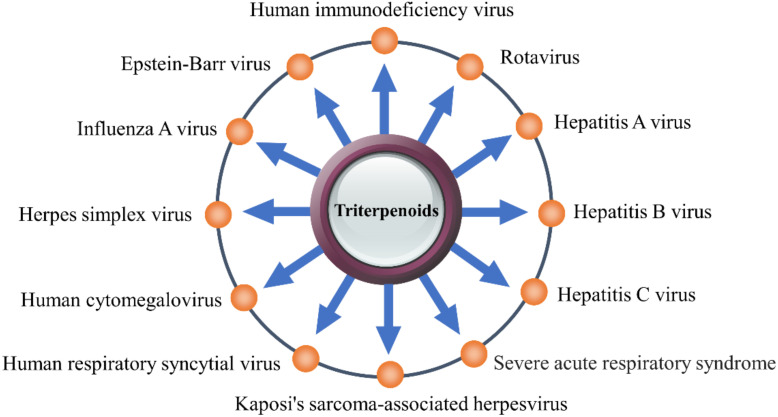
The triterpenoids have inhibitory effects on a variety of viruses.

**Figure 2 pharmaceuticals-15-01169-f002:**
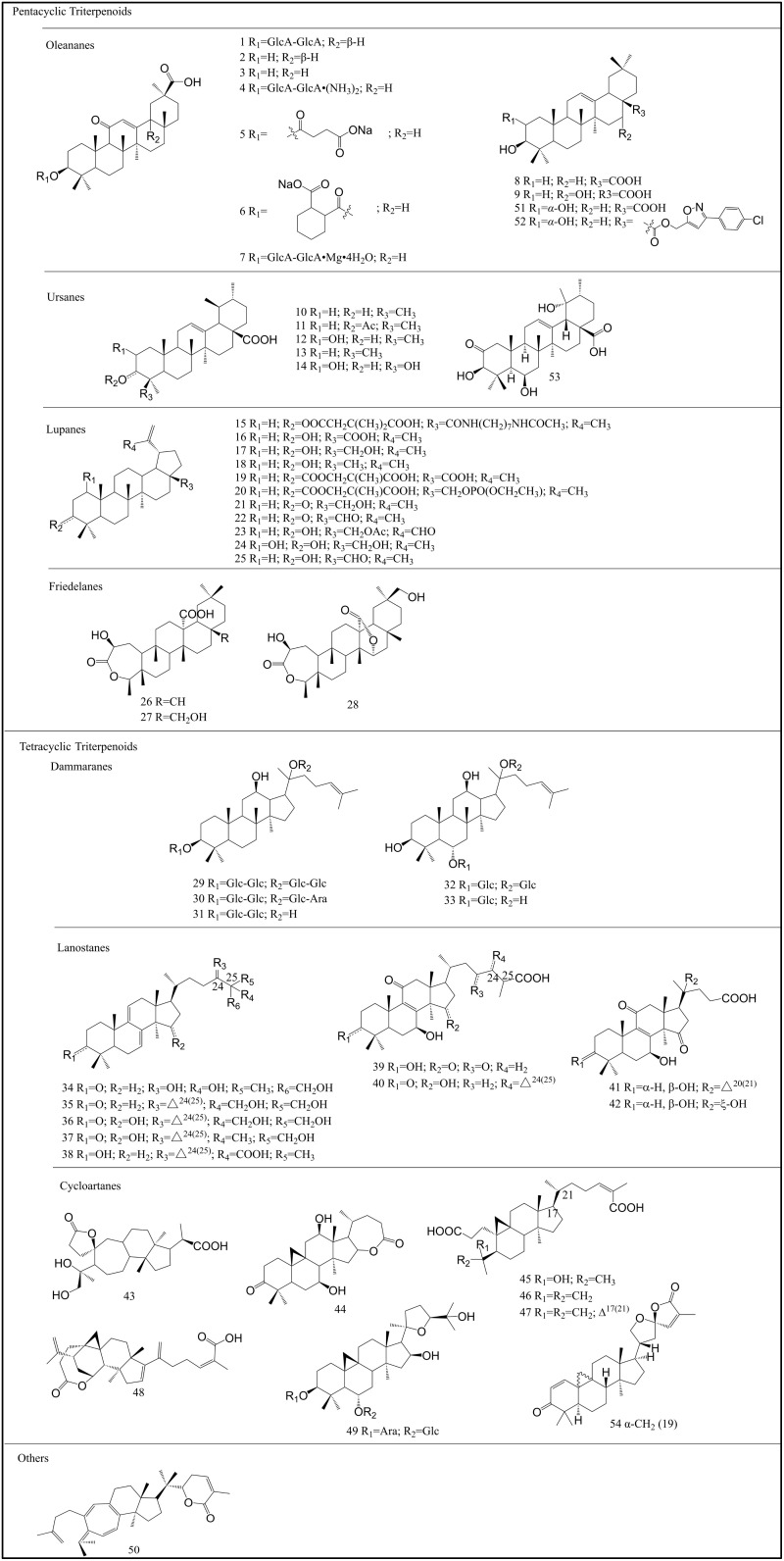
Different types of triterpenoids with antiviral activities.

**Figure 3 pharmaceuticals-15-01169-f003:**
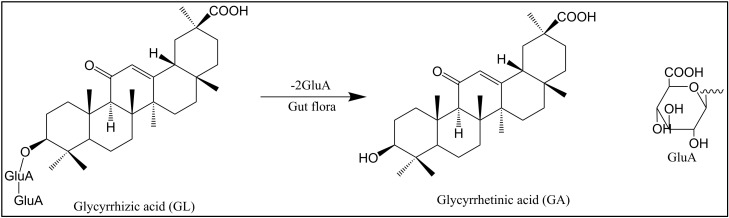
Chemical structures of glycyrrhizic acid (GL) and glycyrrhetinic acid (GA).

**Figure 4 pharmaceuticals-15-01169-f004:**
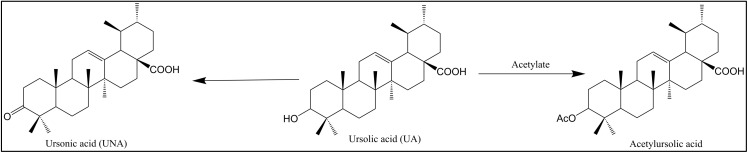
Chemical structures of ursonic acid (UNA), ursolic acid (UA), and acetylursolic acid.

**Figure 5 pharmaceuticals-15-01169-f005:**
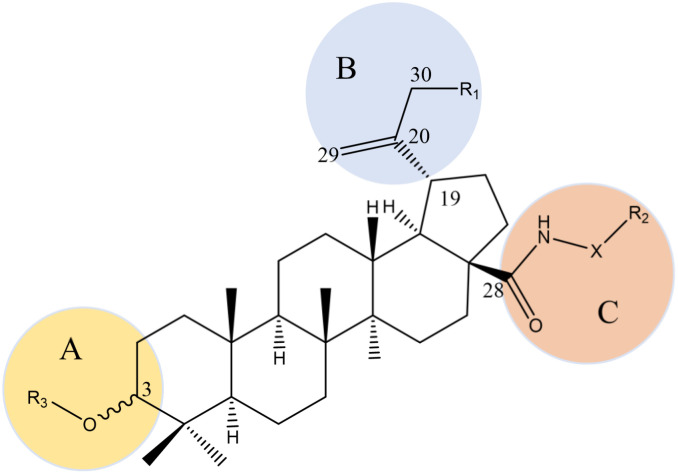
Structure–activity relationships of BA derivatives against HIV. **A:** 1. Activity disappears after introduction of fluorine at the C-2 position, and the proton at the C-2 position may play a key role in anti-HIV activity. 2. 3-β configuration is better than 3-α configuration. 3. The position of the dimethyl substitution on the side chain greatly influences activity. 4.The activity of the C-3 ester group is replaced by the bioisostere amide. **B:** 1. The isopropenyl group at position C-19 may not be an active pharmacophore, but proper modification can improve the water solubility of the drug, and thereby improving the pharmacokinetic properties of the drug. 2. R_1_ as a hydrogen bond donor is unfavorable to the activity. 3. Changing the C-30 allyl group has no obvious effect on drug activity. **C:** 1. C-3 acylation can inhibit HIV maturation and C-28 amidation can prevent HIV from integrating into cell membranes. 2. The atomic length of the side chain at position C-28 is extremely significant for improving the antiviral efficacy, and the side chain structure containing amide groups is very important for activity. 3. The free hydroxyl group of R_3_ and the free carboxyl group at the end of R_2_ are necessary for anti-HIV activity. 4. When X is a secondary amine ring such as piperidine, it can significantly improve metabolic stability.

**Figure 6 pharmaceuticals-15-01169-f006:**
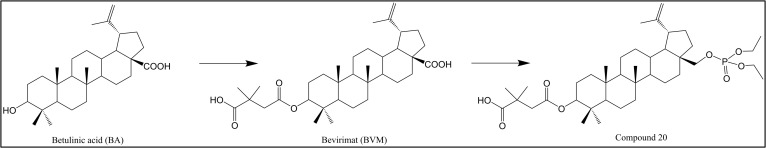
Chemical structures of betulinic acid (BA), bevirimat (BVM), and compound **20**.

**Figure 7 pharmaceuticals-15-01169-f007:**
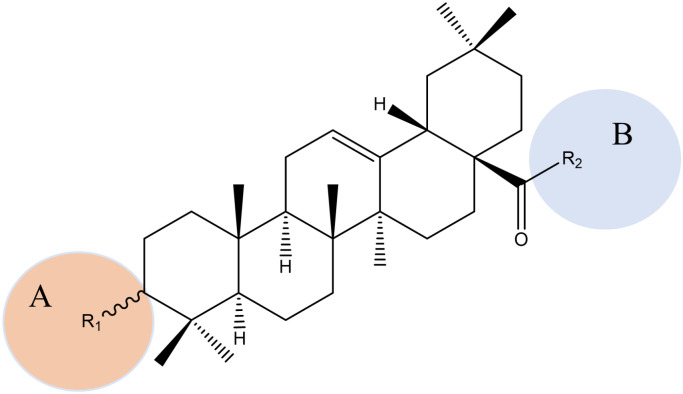
Structure–activity relationships of OA derivatives against HCV. **A:** 1. 3-R1 in the β configuration is better than α configuration. 2. A free carboxyl group at the end of R1 is better than a free amino group. **B:** The R2 hydroxyl group is better than a methoxy group, and has the potential to inhibit HCV protease.

**Table 4 pharmaceuticals-15-01169-t004:** Summary of anti-HAV triterpenoids and their targets and activities.

Compounds	Targets	Cells	References
**1**	Replication inhibitors	PLC/PRF/5 cells	[78]
**29; 32**	-	Fetal rhesus monkey kidney (FRhK-4) cells	[79]

**Table 7 pharmaceuticals-15-01169-t007:** Summary of anti-RV triterpenoids and their targets and activities.

Compounds	Targets	Cells	References
**10**	Replication inhibitors	MA104 cells	[92]
**12; 30; 31**	Immunomodulators	MA104 cells	[91,93]

**Table 10 pharmaceuticals-15-01169-t010:** Summary of anti-EV71 triterpenoids and their targets and activities.

Compounds	Targets	References
**1; 8; 10; 14**	Replication inhibitors	[87,119]
**29**	Immunomodulator	[118]
**37; 38**	Entry inhibitors	[120]

## Data Availability

Data is contained within the article.

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
