# Peer review of "Recent Advances in Antiviral Activities of Triterpenoids"

_pharmaceuticals, 2022, doi:10.3390/ph15101169_

Round 1
Reviewer 1 Report
Generally, the proposed manuscript is well-organized and well-written. Antiviral drugs are now one of the most critical topics. Although many papers focus on this topic, the discussion about tritenteoins and their application is a new and fresh look.
Before publication, I recommended small changes and corrections:
Line 38-42, please added information that vaccination is not effective in 100%, so still, even vaccinated people have "chances to catch the virus."
Figure 1 is not informative for general readers; please do not use acronyms in this place.
Additionally, the authors do not quote some limitations of triterpenoids which significantly limit their medical use. Generally, triterpenoids are highly lipophilic substances ( for example, https://doi.org/10.1016/j.chroma.2021.462552).
Also, its bioavailability is usually very low. These obstacles should also be presented together with a narrative discussion about the potential conception of its solutions.
Author Response
Response to Reviewer 1 Comments
Generally, the proposed manuscript is well-organized and well-written. Antiviral drugs are now one of the most critical topics. Although many papers focus on this topic, the discussion about tritenteoins and their application is a new and fresh look.
Before publication, I recommended small changes and corrections:
Line 38-42, please added information that vaccination is not effective in 100%, so still, even vaccinated people have "chances to catch the virus."
Response: Thanks for the reviewer's valuable suggestions. We have added the information
as suggested by the reviewers in the revised manuscript.
Figure 1 is not informative for general readers; please do not use acronyms in this place.
Response: Thanks for the reviewer's valuable suggestions. We have changed the acronyms of viruses into full names in the revised manuscript.
Additionally, the authors do not quote some limitations of triterpenoids which significantly limit their medical use. Generally, triterpenoids are highly lipophilic substances (for example,
https://doi.org/10.1016/j.chroma.2021.462552). Also, its bioavailability is usually very low. These obstacles should also be presented together with a narrative discussion about the potential conception of its solutions.
Response: Thanks for the reviewer's valuable suggestions. We have supplemented some contents of limitations of triterpenoids, and we discussed the reasons for the low bioavailability of triterpenoids and proposed possible solutions in the discussion section.

Reviewer 2 Report
Summary of the key contribution of the paper:
Recent Advances in Antiviral Activities of Triterpenoids review gives an overview and a discussion of triterpenoids as potential antiviral agents. The present review provide evidence to support the effectiveness of the strategy of discovering new antiviral drugs from triterpenoids.
Highlights:
· Review clearly articulates the use a discussion of triterpenoids as potential antiviral agents in a wide variety of medical applications
· Figures and tables are well referenced and clear
· Antiviral Activities application and research is up to date
Lowlights:
· The bulk of this review (with exception of Antiviral Activities applications) focuses on studies done between 2000-2015. It would be helpful to include more recent studies, especially given the use of triterpenoids for treatment of COVID-19.
The following criticisms are raised for the authors to address in their revision.
Q1: Methodology explanation about HIV on page 10 needs more detail. A flow chart showing the initial number of results, number of papers excluded, and final number evaluated would be helpful.
Q2: Page 5 lines 66-69 should have a reference
Q3: The application of triterpenoids in HIV treatment is heavy on early 2000s research with little information from the past 4 years. Are there more studies that could be included from 2018-2022?
Q4: Page 39 conclusion should be developed more to discuss possible future applications of the suggested modified triterpenoids.
Author Response
Response to Reviewer 2 Comments
Recent Advances in Antiviral Activities of Triterpenoids review gives an overview and a discussion of triterpenoids as potential antiviral agents. The present review provide evidence to support the effectiveness of the strategy of discovering new antiviral drugs from triterpenoids.
The following criticisms are raised for the authors to address in their revision.
Q1: Methodology explanation about HIV on page 10 needs more detail. A flow chart showing the initial number of results, number of papers excluded, and final number evaluated would be helpful.
Response: Thanks for the reviewer's valuable suggestions. In the revised manuscript, we have added the flow chart as the reviewer suggested.
Q2: Page 5 lines 66-69 should have a reference
Response: Thanks for the reviewer's valuable suggestions. In the revised manuscript, we have added the reference as the reviewer suggested.
Q3: The application of triterpenoids in HIV treatment is heavy on early 2000s research with little information from the past 4 years. Are there more studies that could be included from 2018-2022?
Response: Thanks for the reviewer's valuable suggestions. we have updated the research of triterpenoids against HIV in the past five years.
Q4: Page 39 conclusion should be developed more to discuss possible future applications of the suggested modified triterpenoids.
Response: Thanks for the reviewer's valuable suggestions. We have supplemented some contents of limitations of triterpenoids, and we discussed the reasons for the low bioavailability of triterpenoids and proposed possible solutions in the discussion section.
